# Important Aspects Influencing Delivery of Serious News in Pediatric Oncology: A Scoping Review

**DOI:** 10.3390/children8020166

**Published:** 2021-02-22

**Authors:** Lucie Hrdlickova, Kristyna Polakova, Martin Loucka

**Affiliations:** 1Department of Pediatric Hematology and Oncology, University Hospital Motol, 156 00 Prague, Czech Republic; 2Pediatric Supportive and Palliative Care Team, University Hospital Motol, 156 00 Prague, Czech Republic; 3First Faculty of Medicine, Charles University, 121 08 Prague, Czech Republic; 4Center for Palliative Care, 101 00 Prague, Czech Republic; k.polakova@paliativnicentrum.cz (K.P.); m.loucka@paliativnicentrum.cz (M.L.); 5Third Faculty of Medicine, Charles University, 100 00 Prague, Czech Republic

**Keywords:** communication, serious news, pediatric oncology

## Abstract

Delivering serious news presents a major challenge for clinical practice in pediatric oncology due to the complexity of the communication process and a number of aspects that influence how the serious news is delivered and received. This study aims to review and explore the aspects influencing the delivery of serious news in pediatric oncology from the perspective of physicians, parents, siblings and patients themselves. The MEDLINE, Embase, Scopus, Cochrane Library, PsycInfo and Medvik databases were systematically searched for relevant articles published from 1990 to 2017. Following the Preferred Reporting Items for Systematic Review and Meta-analysis extension for scoping reviews (PRISMA-ScR) guidelines, 36 original papers were included. Identified aspects of communication were categorized into six thematic groups: initial setting, physician’s approach, information exchange, parental role, illness related aspects and age of the ill child. The importance of the aspects is perceived differently by parents, patients, siblings and physicians. This scoping review highlights that delivering serious news requires an individualized approach towards the patient and the family. Ten key objectives built upon the results of the literature review offer guidance for daily clinical practice in communication with pediatric patients and their families.

## 1. Introduction

Communication is an essential tool for building trust and alliance with seriously ill children and their families [1,2,3,4]. In pediatric oncology, delivering serious news represents one of the biggest challenges for physicians, although an honest approach and open communication can improve patients’ adherence to treatment [5]. Children who receive information about their diagnosis and prognosis during the initial stage of the disease are less anxious and less depressed than children who receive the information at a later stage [6]. Studies have shown that most parents wish to be fully informed about their child’s diagnosis and prognosis [5,7] and so do the majority of the patients [8,9]. Effective communication facilitates advanced care planning and end-of-life discussions [7,10,11]. Informing parents about their child’s incurable cancer in an appropriate manner also increases the chance that they will better absorb the information about their child’s impending death [12].

Despite the current trend towards open and honest communication [13], healthcare professionals still lack clear guidance on how better communication might be actually achieved [14,15]. The urgent need for evidence-based guidelines to navigate pediatricians in providing effective communication with patients and their families has been repeatedly highlighted in the literature in the last decade [3,16]. Efforts to establish guidelines for clinical practice in pediatric oncology have been built upon successful initiatives from adult oncology and palliative medicine, such as the widely acknowledged SPIKES protocol [17,18,19,20]. Communication issues, especially the ones regarding end-of-life discussions and maintaining hope, have become one of the crucial topics in pediatric palliative care [4,21]. However, communication with children and families at the end of life still remains a great challenge [7,15]. Identification of aspects of the communication process during the delivery of serious news presents the first step for pediatric oncologists to understand the unique perspective and various preferences of patients and their family members.

## 2. Methods

### 2.1. Aim

The objective of this scoping review was to identify aspects influencing communication about serious news in children and adolescents diagnosed with cancer from the perspective of patients, their families and the care-providing physicians.

### 2.2. Design

We conducted a scoping review in order to answer the research question. Scoping reviews are particularly useful when mapping the key concepts of a studied phenomenon and bringing together evidence from heterogeneous literature sources [22]. To ensure methodological rigor, we followed the Preferred Reporting Items for Systematic Review and Meta-analysis extension for scoping reviews (PRISMA-ScR) [23].

### 2.3. Inclusion Criteria

Qualitative and quantitative papers which reported original research findings, published between 1990 and 2017 in English, French and Czech language and focused on children and adolescents with cancer up to 18 years of age, their family members (parents and siblings) and/or physicians providing care to pediatric oncology patients were surveyed in this scoping review.

With the aim to narrow down the complex issue of communication, four specific situations were selected, representing the most challenging moments during the treatment of childhood cancer—initial diagnosis, relapse, end of curative treatment and end-of-life care. These situations reflect the breaking points when serious decisions regarding the future care must be made and difficult conversations often take place. Only studies reporting on communication around these issues were included in this review.

We excluded reviews, perspective papers and commentaries, editorials and letters to editor, although their references were checked for additional, previously non-identified articles relevant to this review. Studies which reported on communication in pediatric oncology but did not include any of the four communication points as stated above were also excluded.

### 2.4. Information Sources

All available articles in English, French and Czech and indexed in the following databases were searched: MEDLINE, Embase, Scopus, Cochrane Library, Medvik and PsycInfo.

### 2.5. Search

Identification of the studies was based on an electronic database search using the search terms listed below. The electronic search strategy itself was divided into three steps:Preliminary search in the MEDLINE database to identify relevant keywords using MeSH terms. The initial search terms were:
“child”“tumor” OR “tumour”“communication”Keywords and terms identified through this preliminary search were used for the extensive search of the literature. For the MEDLINE database, the following formula was used: (child* OR pediatric* OR paediatric*) AND (oncolog* OR tumor OR tumour* OR cancer OR neoplasm*) AND communicat*.Reference lists and bibliographies of the manuscripts retrieved from stages (1) and (2) were searched.

The search formula is available in the Appendix A.

### 2.6. Study Selection

All papers identified through the searching process were processed in the bibliographical management tool Mendeley. Duplicates were eliminated both electronically and manually. We screened all articles by title and abstract, which was followed by reading articles identified as relevant in full text by two researchers (L.H. and K.P.). Any disagreements were resolved via discussion and a third reviewer (M.L.). A PRISMA flow diagram (Figure 1) and a PRISMA checklist (Appendix A) were created.

### 2.7. Data Extraction and Analysis

Data from studies included in this review were extracted into an Excel spreadsheet designed for this purpose and divided into four categories based on the study participants: physicians, patients, parents and siblings. Data extracted from the studies were analyzed through an inductive and data-driven coding process following Brown and Clark’s approach of thematic analysis [24] by two researchers (L.H. and K.P.). Identified codes were sorted and collated into candidate themes followed by re-reading of the texts, adjusting the identified themes accordingly and developing final thematic categories.

## 3. Results

The literature search identified 1405 citations. After removal of duplicates, 987 citations were examined for eligibility by screening the title and abstracts, resulting in 134 papers retrieved in full text. Based on the inclusion criteria, 98 papers were excluded. In total, 36 papers were included for further analysis. This process, including the reasons for exclusion, can be seen in the PRISMA flow diagram (Figure 1).

In total, 22 aspects of communication were identified in the reviewed literature. Aspects of the communication process in delivering serious news are summarized in Table 1. Identified aspects were divided into six thematic groups (Table 1).

### 3.1. Study Characteristics

The methodological design of the included studies was heterogeneous, with twenty-one studies being qualitative, eleven being quantitative and four used a mixed methods design. Most studies explored parental preferences regarding delivery of serious news in pediatric oncology, referring either exclusively to parents or exploring combined preferences of parents and patients/physicians. There were four studies focusing exclusively on the perspective of physicians and other medical staff members. Only two studies explored aspects influencing serious communication with siblings of patients. The 36 studies included in our review present data from 352 patients, 2257 family members and 931 physicians. The included papers were from several countries, with eight studies from the United Kingdom, seven from the United Stated and four studies each were from The Netherlands and Sweden. Canada and Switzerland were represented by two studies each and Ireland, France, Belgium, Slovakia, Iran, Malaysia, Australia, Japan and the Republic of South Africa by one study each. The design and methods of all reviewed articles are specified in Table 2.

### 3.2. Thematic Groups

The initial coding process led to the identification of 94 codes spread across four categories of participants (Appendix A).

The identified themes are the following: initial setting, physician’s approach, information exchange and parental role. Illness-related aspects and patient’s age were considered as two separate thematic groups. Different aspects regarding the delivery of serious news were reported by different participant categories (physicians, parents, siblings and patients). Only two aspects were reported as significant by all four categories of participants: patient’s age and respect.

Physicians, parents and patients all emphasized honesty and empathy as important aspects of the communication process. Most aspects identified by patients and parents were identical. Some aspects emphasized by parents were not identified in the category of physicians—the initial setting and the information exchange in particular. Parents emphasized privacy, the importance of the vocabulary used, the amount of information delivered and the clarity of information.

Two unique aspects were mentioned by physicians: responsibility and communication training. The importance of professional expertise, personal responsibility and the need for training in communication skills are crucial aspects of communication for physicians.

In the category of siblings, only three aspects were identified: patient’s age, respect and the amount of information delivered.

Illness-related aspects were determined only in the category of parents and physicians. Clearly, the diagnosis and the seriousness of the disease are not strongly important aspects of the communication process for patients. Thematic groups based on the identified aspects are discussed in detail below.

### 3.3. Initial Setting

Aspects related to the initial setting of how serious news is delivered were identified predominantly in studies focused on the parental perspective. Parents stressed privacy [30,37,43]. The location of the conversation was less important if privacy was ensured [30]. In contrast, parents described their experience as disruptive when they felt that their privacy was violated by others [30,43]. Another issue identified by parents was related to who is present when the serious news is delivered, including the patient [5,33,51,55], other family members [30,43,44] and medical staff [30,44]. In some studies, parents preferred the child not to be present during diagnosis disclosure as this is an overwhelming situation and parents did not want to show any weakness in front of their child [30,44,55]. Parents’ perception of the patient’s presence was influenced by the child’s age, which showed up to be an independent aspect. Another important aspect highlighted by parents was the timing and duration of the information delivery. Most parents wanted to be given accurate information about their child’s cancer as soon as possible [12,26,49] and they needed to have enough time for the consultation [30,33,44]. The perception of how much time was enough for delivering serious news was very individual and varied between 30 and 90 min [30]. Parents needed enough time to be able to come to terms with the disturbing information, to be able to ask questions and to express their emotions [30,33,44].

Physicians acknowledged the need for having enough time and being well prepared for the meeting with parents while delivering serious news to them [32].

The only aspect highlighted by patients in the initial setting group was their presence during the diagnosis disclosure, but their opinion varied greatly from being told the serious news before parents were informed [48] or being informed together with parents [25,33] to being informed later on [33,55]. This aspect is influenced greatly by the age of the patient, with older children and adolescents wanting to be involved more than younger children [25,48].

### 3.4. Physician’s Approach

The approach of physicians towards parents and patients represented a key theme associated with delivering serious news and was mentioned by all participants. All four categories of participants reported physician’s respect as an important aspect of communication. Parents wanted to be respected and taken seriously by their child’s physician [26,31,33]. Being treated respectfully was also very important to the patients and their siblings [31,41,48]. Physicians highlighted the fact that treating patients and families respectfully had a great influence on all serious conversations [55]. Other aspects frequently mentioned by parents, physicians and patients were empathy [7,9,29,31,36,43,45,46,47] and honesty [25,31,33,43,44,53,54]. Parents valued an empathetic approach of physicians while delivering serious news, which allowed them to express their emotions and receive support from the medical staff [7,9,29,45]. The lack of empathy and its negative impact on parental experience was mentioned in some of the studies [44,49,56]. Parents emphasized the need for physicians to be more human and understanding [36] as their behavior was perceived sometimes even as aggressive and rough [44]. Patients also valued physicians’ empathic approach [9,45,48]. Adding to this, patients valued open and honest communication as this made them feel included in the communication process [25,33,45,48]. An honest and open approach was also highly valued by parents, even if the disclosed information was upsetting or shocking [25,31,33,43,44,45,54]. Parents valued an honest delivery of information if done with compassion [34,45] and appreciated if physicians honestly acknowledged not being certain about the outcome [33]. The need for an open and honest approach was also identified in studies with healthcare providers [25,32,39,41]. Another identified aspect was maintaining hope, even in adverse situations [5,25]. Maintaining hope represented a coping strategy for mothers of children with cancer as it helped them to adapt to the serious news [44]. A study set in Japan identified a “never-give-up attitude” which should be present until the end, although parents in this study wanted to have adequate information regarding worst-case scenarios at the same time [45].

Cultural background and its impact on the physician’s approach presents an important issue in diagnosis disclosure and other difficult conversations. Doctors from the U.S. and western Europe tend to deliver serious news openly to both the parents and the children [25,35,39,41]. In Asian and African countries, the understanding of cancer in children is different, and that might result in different approaches of physicians and parents; parents report understanding that cancer is an illness of elderly people [40], so cancer in children might be perceived as a “death sentence” [28] or they might think of cancer as of their child’s spell [43]. In Asian and African countries, physicians are often asked by parents to avoid some specific “brutal” vocabulary (e.g., cancer or chemotherapy) [43,44]. With regard to the current demographic trends and multicultural background of patients, sensitivity to cultural differences represents an important clinical task.

### 3.5. Information Exchange

Information exchange was emphasized by both parents and patients. Both participants highlighted the need for using appropriate language and vocabulary without medical jargon [6,26,28,33,44]. Included studies indicated that using the word “cancer” could be perceived by parents with different feelings because of its negative connection with death [25,28,44]. The clarity of information delivered by physicians during the delivery of bad news was another important aspect of communication. Parents wanted to have clear information about the illness and treatment, which included the opportunity to ask questions to clarify confusing issues and make informed decisions [31,33]. However, one study showed that 30% of parents participating in the research did not understand the content of the message [43].

Providing the right amount of information was mentioned in several studies [5,6,26,27,30,31,33,45,49,53]. In several studies, parents reported being so shocked and stressed about the information disclosure that it prevented them from listening to all the information delivered by the physicians [26,27,33,49,53]. Parents therefore emphasized their need to have the information delivered repeatedly [31,33,45,49,53]. Several studies reported parental preferences to receive as much information as possible during the diagnostic and prognostic disclosure [5,30] and to be fully informed about the worst-case scenario [45,54], while in other studies, parents preferred to be given only basic information at the time of initial diagnosis and wanted further details later [33,49,53]. Parents and patients also desired to have other information sources when needed (booklets, brochures and physician’s availability for questions) [26,30,33,37,49].

Similar to their parents, patients desired to be told the right amount of information reflecting their preference [25,33,55] and the use of words they could understand [33,45]. Based on the only two included studies, siblings wanted to be informed in detail about their brother’s or sister’s illness [31,38].

Physicians emphasized their professional expertise, responsibility and the need for communication training as the important aspects during the delivery of serious news to patients and families [25,32,35,50].

### 3.6. Parental Role

Other aspects which influenced the delivery of bad news were connected to the role of parents. Parents wanted to protect their child by limiting the amount of information they receive, especially if the child was emotionally unstable or was of a younger age and would not be able to cope if the information were disclosed in full [6,25,27,33]. Parents perceived themselves as experts on their child’s personality, and therefore, they usually felt like the most competent to set the boundaries on information disclosure [25]; however, they welcomed support provided by the medical professionals while talking to the child [42]. Respecting the needs of parents who may desire to withhold information from the child presented a challenge for the physicians, as this put them in difficult position [27,41]. Another parental role was to be an advocate for their child [25,33,55].

### 3.7. Patient’s Age

An important aspect which affected the delivery of serious news was the age of the ill child and the severity of the diagnosis. The importance of the child’s age was mentioned in several studies by parents [6,9,25,28,30,33,34,44,51,52], physicians [32,35], patients themselves [9,33,40,55] and even siblings [38]. Patient’s age affected how much information would be delivered to them and who would the news be delivered by. Parents preferred to deliver serious news to younger children by themselves [9,25]. At the same time, the younger the child was, the less information was likely to be told [6,9,25,28,30,40,44,52]. Adolescents often preferred to join the diagnosis disclosure or other situations of delivering serious news and they wanted to make decisions by themselves [9,33,41,48,55]. However, for some adolescents, parents were welcomed to act as their advocates [25,33,55]. When treating adolescents, physicians sometimes described a conflict of interest, as their loyalties were torn between the patients and their parents, especially when parents asked the physician to withhold information from the adolescent patient [41].

### 3.8. Illness-Related Aspects

Severity of the diagnosis, poor prognosis and other aspects related to the illness were categorized as a separate, independent group of aspects. The more severe the illness, the less often the child was present during diagnostic and prognostic disclosure [12], and parents would ask physicians to approach their child with optimism [28], but sometimes, the nature of the diagnosis (tumor or leukemia), and the prognosis (good or poor), did not have any effect on the way parents gave information about the disease to their child [6].

## 4. Discussion

This scoping review identified aspects influencing communication during the most challenging moments in the treatment of childhood cancer: the initial diagnosis, relapse, end of curative treatment and the end-of-life care, from the perspective of parents, patients, physicians and siblings. Identification of these aspects presents a complex framework, useful for pediatric oncologists navigating the process of communicating with children with cancer and their families.

Our study reveals the difference between parental, patients’ and physicians’ perspectives in leading difficult discussions when serious news is delivered. The results show that while parents’ and patients’ perspectives are very much alike, a striking difference was identified in physicians’ perspectives. Parents and patients both emphasized factors related to the information exchange itself: the vocabulary used, the amount of information delivered, the clarity of information and their need to have written sources of information. In contrast, physicians emphasized their responsibility of being the messenger of the serious news, their professional expertise and the need of training in communication skills. The lack of training has been previously stressed by pediatric oncologists elsewhere [15,50,56]. Lack of communication training opportunities and insufficient recommendations for effective communication in pediatric medicine increase physicians’ uncertainty in how effective communication might be actually achieved [14,15,16,32,50]. Guidelines for sharing life-altering information in pediatrics as a modification of the famous SPIKES framework were presented in 2014 [57]. However, there is still limited evidence about the feasibility and effectiveness of adopting adult-based guidelines in the pediatric setting [57,58]. Developing guidelines for pediatric oncology cannot be just an extension of SPIKES but must be based on the complexity and specificity of situations of families whose child is treated with cancer. Recently, key communication skills were addressed to improve effective communication and promote therapeutic alliance in the clinical practice of pediatric oncologists [59]. Communication training seems to be a crucial tool for improving communication with patients and their families [50,58]. Although there are training models available that are proven to be effective in improving communication skills [20,50], they are not broadly implemented in pediatric fellowship training curricula yet [60].

A landmark historical study by Sisk et al. [13] analyzed the changes in the communication with pediatric patients with serious illness over recent decades. Our review supports their results by identifying older publications reporting parental reluctance to be informed in detail [52], contrasting with the more recent trend to receive as much information as possible [5]. At the same time, parental protectiveness seems to remain similar across the decades. Papers from the 1990s as well as the recent papers often mention parents’ desire to keep their child unaware of the truth or in mutual pretense [6,25,33,39,42,52]. The historical analysis must take into account the significant progress in the curability of pediatric cancers in the last 30 years, which might have influenced parental expectations and their preferences to discuss the disease openly.

Our findings support the need for an individual approach toward the patient and the family based on eliciting patients’ and family members’ preferences. Asking patients and their families for permission before informing them would be one of the initial steps recommended in serious news delivery guidelines in the adult patient population [17] and in the newly adapted model for pediatrics too [58]. As You et al. highlighted in their work [61], physicians tend to worry that the patient or the family are not ready for the bad news, and this leads to postponing difficult discussions. However, clinicians do not always accurately perceive patients’ emotions and their satisfaction with communication [62]. Continuous assessment of the information perception, allowing to deliver tailored information according to the individual preference, can possibly meet parental and patients’ information needs; however, this is clearly an area for further research.

### 4.1. Clinical Implications

Based on the aspects identified in our scoping review, 10 key objectives were formulated to help understand the principles of delivering serious news in pediatric oncology. The key objectives present 10 recommendations to be acknowledged and adopted by pediatric oncologists before accepting the challenge of delivering serious news to a patient and/or family. The key objectives can serve as a foundation for standardized guidance for effective communication, not only with pediatric cancer patients and their families but for pediatric clinical practice in general (Table 3).

### 4.2. Strength and Limitations of the Study

This scoping review has several strengths. The literature search was conducted in major databases and the comprehensive search strategy was designed by a qualified librarian. Our review includes papers from five continents and therefore brings a comprehensive perspective of patients, families and physicians from all over the world, which is especially important in the multicultural environment of the 21st century that we are facing in clinical practice. The data from the included studies were extracted and analyzed by two researchers following the PRISMA-ScR guidelines to meet the standards for this kind of literature review.

There are several limitations of this review. We only included studies published in English, French and Czech; thus, we could have missed important evidence written in different languages. By using a wide time range for the search, we also included studies from the early 1990s, where the communication about diagnosis could have been influenced by significantly worse prognosis in many diagnostic groups than it is nowadays, especially from the physicians’ point of view. Despite this fact, we believe that parental and patients’ perspectives remain very much the same, as cancer is still understood as a very serious and life-threatening illness.

In our scoping review, papers with physicians’ perspective on delivering serious news were included. Further research shall also focus on understanding the communication preferences and practices of non-physician members of the clinical team whose perspective also plays an important role.

Although our review included the category of siblings, we only identified two studies exploring factors influencing serious discussions related to siblings. Our findings regarding this group of participants are therefore limited.

This review aimed to identify the aspects that influence communication about serious news with patients and their families. Further analysis of specific aspects, their value or direction (positive/negative) could provide more information about the specific role of these aspects.

## 5. Conclusions

Numerous aspects of communication were identified as important for delivering serious news in pediatric oncology. Physicians’ understanding of a patient’s and parental perspective plays a crucial role in determining an appropriate way of delivering serious news. An individualized approach, therefore, appears to be the fundamental axiom for effective communication of serious news.

Bearing in mind the large number of identified aspects of communication about serious news in pediatric oncology as well as the lack of confidence and training in communication skills, several issues must be addressed in future research. Further research is needed to address specific parental and patient needs regarding the delivery of serious news in pediatric oncology in specific cultural settings. More research is needed to understand siblings’ preferences in communication about serious issues.

The available guidelines must be reviewed and tested for their effectiveness and implemented in education and clinical practice. Better understanding of the cultural context which shapes the needs and preferences of patients and their families is also necessary.

## Figures and Tables

**Figure 1 children-08-00166-f001:**
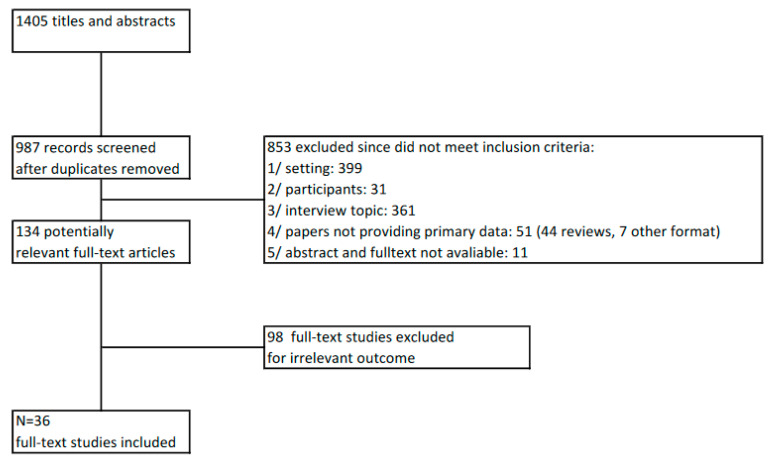
Preferred Reporting Items for Systematic Review and Meta-analysis (PRISMA) diagram.

**Table 1 children-08-00166-t001:** Aspects of the communication process in delivering serious news.

Group No.	Thematic Group	Aspect		Participant Category			
				Parent	Physician	Patient	Sibling
I	Initial setting	1	Privacy	x			
		2	Patient’s presence	x		X	
		3	Family member’s presence	x			
		4	Staff member’s presence	x	x		
		5	Time and timing	x	x		
II	Physician’s approach	6	Empathy	x	x	X	
		7	Honesty	x	x	X	
		8	Respect	x	x	X	x
		9	“Never-give-up attitude”	x			
		10	Close contact	x	x		
		11	Responsibility		x		
III	Information exchange	12	Language and vocabulary	x		X	
		13	Clarity of information	x		X	
		14	Amount of information	x		X	x
		15	Information content	x			
		16	Access to information	x		X	
		17	Training		x		
IV	Parental role	18	Child-protective role	x	x		
		19	Advocacy		x	X	
		20	Trust	x	x		
V	Illness-related factors	21		x	x		
VI	Age	22		x	x	X	x

**Table 2 children-08-00166-t002:** Characteristics of the included studies.

Reference	Study Design	Study Objective	Country	Participants	Identified Aspects of Communication
I Coyne, 2016 [25]	Qualitative, audio-recorded individual interviews	To examine participants’ views on children’s participation in decision sharing and communication interactions	Ireland	20 patients, 22 parents	I–IV, VI
JM Snaman, 2016 [7]	Qualitative, tape-recorded discussions in focus groups	To explore communication between staff members and patients and families near the end of life	USA	22 bereaved parents	II
JE Maree, 2016 [26]	Qualitative individual interviews	To explore information needs of parents of children with cancer in South Africa	South Africa	13 parents	I–III
DO Badarau, 2015 [27]	Qualitative, individual semi-structured interviews	Identification of factors that contribute to restricted provision of information about diagnosis to children in Romania	Romania	18 parents, 10 oncologists	III
A Watanabe, 2014 [28]	Qualitative, individual semi-structured tape-recorded interviews	To examine parents’ and grandparents’ views on deciding to share or not to share the diagnosis with the adolescent	Japan	55 parents, 3 grand-parents	III, V, VI
B Young, 2013 [29]	Qualitative, audio-recorded individual semi-structured interviews	Examine perspectives of parents of children with acute lymphoblastic leukemia on discussing emotions with oncologists	UK	67 parents	II
RM Kessel, 2013 [30]	Cross-sectional study, mixed design, questionnaire designed by expert panel	Parental preferences when receiving information about child’s diagnosis (setting, length and other parameters of the interview)	USA	62 parents	I, III, VI
AC Steele, 2013 [31]	Cross-sectional qualitative study, individual interviews with open-ended questions tape-recorded	How to improve care for families during the end of life	Canada	36 mothers, 24 fathers, 39 siblings	II, III
M Zwaanswijk, 2011 [9]	Experimental mixed design, questionnaire	Investigate preferences of children with cancer, their parents and survivors regarding medical communication	The Netherlands	34 patients, 59 parents, 51 survivors	II, VI
M Stemarker, 2010 [32]	Qualitative, tape-recorded semi-structured interviews	Main concerns of physicians when informing about malignant disease and psychosocial aspects of being pediatric oncologist	Sweden	10 oncologists	I–III, VI
P Lannen, 2010 [12]	Quantitative, questionnaire	To assess parents‘ ability to absorb information about their child’s cancer being incurable and to identify factors associated with parents’ ability to comprehend this information	Sweden	191 bereaved mothers, 251 bereaved fathers	I, V
M Zwaanswijk, 2007 [33]	Qualitative, online questionnaire for three different focus groups (patients, parents, survivors)	Interpersonal, informational and decisional preferences of parents and patients and survivors	The Netherlands	7 parents, 11 patients, 18 survivors	I-IV, VI
U Valdimarsdóttir, 2007 [34]	Quantitative, questionnaire	Investigation whether care-related factors (i.e., access to information) predicted the timing of parents‘ awareness of child’s impending death	Sweden	449 bereaved parents	I, V, VI
SK Parsons, 2007 [35]	Quantitative, survey questionnaire	Communication practices at diagnosis, difference between disclosure in Japanese and U.S. physicians	USA, Japan	350 U.S. doctors, 365 doctors from Japan	II, V, VI
Mack et al., 2006 [5]	Quantitative, questionnaire	Evaluation of parental preferences for prognostic information about their children with cancer	USA	194 parents, 20 physicians	II, III, V
B Young, 2003 [29]	Qualitative, semi-structured interviews	Management of communication about illness in young people with life-threatening disease and the role of parents in communication process	UK	19 parents, 13 young people	I–IV
RB Levi, 2000 [36]	Qualitative, interviews in focus groups which were designed to include both parents of children enrolled in clinical trial and parents of children not enrolled	To describe retrospective perceptions of parents of the circumstances of their child’s cancer diagnosis and the informed consent process	USA	22 parents	I–III
BF Last et AMH van Veldhuizen, 1996 [6]	Quantitative, questionnaire	To test the hypothesis that being openly informed about the diagnosis and prognosis benefits the emotional well-being of children with cancer	The Netherlands	56 children, 56 mothers, 54 fathers	III–VI
OB Eden, 1994 [37]	Qualitative, structured questionnaire	To assess the receptiveness of parents to information given about their child’s cancer	UK	23 parents	I, III
T Havermans, 1994 [38]	Qualitative, interviews, questionnaire	How children perceive their lives to be affected because of their sibling having cancer	UK	21 siblings	I, III, V
A Goldman, 1993 [39]	Mixed, questionnaire	To discuss whether children dying in hospital had discussed their death with their families and whether any factors in the family appeared to influence dying in hospital setting	UK	39 died children, questioning the staff members	II
CJ Claflin, 1991 [40]	Quantitative, tape-recorded interviews	To address the issue of information disclosure from the child’s perspective	USA	43 patients	VI
S Essig, 2016 [41]	Qualitative, interviews within focus groups	To explore different perspectives on communicating with adolescents with cancer	Switzerland	12 physicians, 18 nurses, 16 survivors, 8 parents	II, IV
Kuan Geok Lan, 2015 [42]	Qualitative, focus group discussions, audio-taped in-depth interviews	To explore parents’ experiences in the end-of-life care of their children and gather their parents‘ views	Malaysia	15 parents of 9 deceased children (8 diagnosis of cancer, 1 Prader–Willi syndrome)	IV
D Lolonga, 2015 [43]	Mixed, semi-structured interviews	To explore the ideal conditions when disclosing diagnosis to parents of children with cancer	African countries	94 parents, 30 healthcare professionals	I–III
F. Aein, 2014 [44]	Qualitative, semi-structured interviews	To explore how mothers in Iran recall receiving information about their child’s diagnosis of cancer	Iran	14 mothers	I–III, VI
S Yoshida, 2014 [45]	Quantitative, multi-center questionnaire survey	To explore the distressing experience of parents of children with intractable cancer	Japan	135 bereaved parents	II, III
AP Greeff, 2014 [46]	Mixed, cross-sectional survey research design, self-report questionnaire	To explore resilience factors associated with family adaptation after child was diagnosed with cancer	Belgium	26 parents, 25 children	II
IMM van der Geest, 2014 [47]	Quantitative, cross-sectional study, questionnaire assessing grief	To explore parents’ perception of the interaction with healthcare professionals and its influence on long-term grief	The Netherlands	89 bereaved parents	II, III
F. Gibson, 2010 [48]	Qualitative, participatory-based techniques of data collection	To explore children’s and young people’s views of cancer care using innovative methods	UK	16 families	I, VI
A Kästel, 2011 [49]	Qualitative, semi-structured interviews	To explore parents’ views on receiving information during childhood cancer care	Sweden	8 families	I–III
L Kersun, 2009 [50]	Quantitative, 12-question web-based survey	To survey recently graduated fellows about their prior training in communication of difficult news	USA	171 fellows	III
TM Parker, 2008 [51]	Quantitative, questionnaire	To assess how parents recall the initial discussion regarding the diagnosis of cancer	Canada	116 parents	I, VI
PM Hughes, 1990 [52]	Qualitative, questionnaire + semi-structured interviews	To assess psychological distresses of parents having child with cancer	UK	18 parents	V, VI
AC Jackson, 2007 [53]	Qualitative, prospective study using a within-group design with repeated measures over time (face-to-face interviews)	To explore coping, adaptation and adjustment in families of a child with brain tumor	Australia	53 parents	II, III
E Gurková, 2014 [54]	Qualitative, semi-structured in-depth interviews	To analyze the parent’s experience when the treatment of their child diagnosed with cancer had failed and the child had died	Slovakia	Bereaved parents: 1 couple, 3 mothers	II, III

**Table 3 children-08-00166-t003:** Ten key objectives.

1	Know the diagnosis and the relevant data.	The more serious the news is, the more difficult the discussion may be. Aspects regarding the illness and prognosis play an important role in the delivery of serious news.
2	Do not overwhelm families with information.	Prepare the key message you need to deliver, keep it short. Some families might ask for more information right away, others will require more information later.
3	Prepare written information (booklets, leaflets, etc.).	Families find very it helpful to receive information in written form to be able to come back to it later.
4	Treat your patients and their families with respect.	Being treated respectfully is one of the crucial preferences of children with cancer and their families.
5	Be authentic. Do not pretend and do not be afraid of truth.	Patients and families do appreciate your openness and honesty.
6	Be aware of parental protectiveness.	Parents may protect their children from serious news, which presents a potential conflict of interest for the physician as many children and adolescents seek for information about their diagnosis and prognosis.
7	Mind the patient’s age and mental development.	These important aspects determine not only the patient’s ability to understand the information, but also the parents’ level of protectiveness.
8	Acknowledge parents’ tendency to hope for the best.	Parents have a unique ability to maintain hope, even if the information given seems hopeless. Parents maintain hope and often want to believe in miracles, no matter what information they received.
9	Be aware of cultural context.	Cultural aspects play an important role in patients’ and parental perspectives.
10	Get used to individualized approach.	Do not assume. What works for one family may not work for another.

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
