# Peer review of "Important Aspects Influencing Delivery of Serious News in Pediatric Oncology: A Scoping Review"

_children, 2021, doi:10.3390/children8020166_

Round 1

Reviewer 1 Report

Review on „Important Aspects Influencing Delivering Serious News in Pediatric Oncology: A Scoping Review”

This is an important comprehensive paper about the communication of stressful events in pediatric oncology from the aspect of everyday clinical practice. The paper is well-written, clear and  easy-to-follow.

Major comments:

It would be interesting to know as the search timeframe is almost 30 years, if there were any differences in communication in different time periods.

The authors mention in Table 3 the importance of „Be aware of cultural context”, but it is not really detailed in the Results section. As the authors declare their ability to overview French and Czech language papers beyond English, and included papers were from several countries, it should have a special interest in this paper.

Minor comments:

  • line 82. The describe what was not searched is unnecessary.
  • line 84: ’manual search’ is described as the part of method, but the way of its use was not clarified
  • Study characteristics section especially from line 130: as it could be seen in the Table I., references sould be removed from here.
  • Table I: this table is too busy, a simplification (use of abbreviation) would be very useful, for Study design and Study objective
  • References should be described according to Instructions for authors

Author Response

Comment 1

It would be interesting to know as the search timeframe is almost 30 years, if there were any differences in communication in different time periods.

Response: Thank you for this helpful observation. An additional information regarding the historical perspective was added in the discussion(lines:394-403).

A landmark historical study by Sisk et al. [13] analyzed the changes in the communication with pediatric patients with serious illness over the last decades. Our review supports their results by identifying older publications reporting parental reluctance to be informed in detail [33], contrasting with the more recent trend to receive as much information as possible [5]. At the same time, parental protectiveness seems to remain similar across the decades. Papers from the 90´s as well as the recent papers often mention parents´ desire to keep the child unaware of the truth or in mutual pretense [6, 25, 33, 35, 41, 53]. The historical analysis must take into account the significant progress in the curability of pediatric cancers in the last 30 years, which might have influenced parental expectations and their preferences to discuss the disease openly.

Comment 2

The authors mention in Table 3 the importance of „Be aware of cultural context”, but it is not really detailed in the Results section. As the authors declare their ability to overview French and Czech language papers beyond English, and included papers were from several countries, it should have a special interest in this paper.

Response: Thank you for this thoughtful comment. A new paragraph about the culture context was added to the result section on page 14 (lines 276-286):

Cultural background and its impact on the physician´s approach presentsan important issue in diagnosis disclosure and other difficult conversations. Doctors from the U.S. and Western Europe tend to deliver serious news openly to both the parents and the children [25, 40, 47, 53]. In Asian and African countries, the understanding of cancer in children is different and that might result in different approach of physicians and parents: parents report understanding that cancer is an illness of elderly people [48], cancer in children might be perceived as “death sentence” [28] or they might think of cancer as of their child´s spell [54]. In Asian and African countries, physicians are often asked by the parents to avoid some specific “brutal” vocabulary (e.g. cancer or chemotherapy) [43, 54]. With regard to the current demographic trends and multicultural background of patients, sensitivity to cultural differences represents an important clinical task.

Comment 3

line 82. The describe what was not searched is unnecessary.

Response: Information about sources not searched was removed.

Comment 4

line 84: ’manual search’ is described as the part of method, but the way of its use was not clarified

Response:By manual search it was meant reviewing the references in the identified papers. Using the term manual was confusing and we deleted it.

Comment 5

Study characteristics section especially from line 130: as it could be seen in the Table I., references should be removed from here.

Response: References were removed from the text.

Comment 6

Table I: this table is too busy, a simplification (use of abbreviation) would be very useful, for Study design and Study objective

Response: We believe that different layout, using a horizontal page display, would help to improve the readability of the Table 1 more than using abbreviations, which would be difficult especially for the study objective column which describes the original aim of each paper in a brief sentence. We will ask the editors whether it is possible to arrange another layout for Table 1.

Comment 7

References should be described according to Instructions for authors

Response: Thank you for precise observation. The references were corrected according to the Instructions for authors.

Reviewer 2 Report

You have clearly done a significant amount of research and completed a comprehensive review of the literature in preparation for submitting this manuscript. You shared that your intended aim was to review and explore aspects that influence communication with specific emphasis on communication during difficult discussions. Table 1 describes the objectives and thematic groupings that you assigned to these studies, but not the individual results of these studies. It is difficult to draw conclusions and make claims without having an understanding of the results of these studies. Table 2 describes certain aspects of the communication process but does not assign a value (positive or negative) to these aspects. Do parents consider the patient's presence good or bad? What about the patients? By just having a check mark in these sections makes me think that parents and patients think it is good to have the patients present, which is not what you are trying to convey. You do discuss this in the results and discussion section, but just viewing this table does not give me a good understanding of the impact of these factors. Being about to provide some assessment of the value of these factors would further support your conclusion that delivering serious news requires an individualized approach. Providing a value to these aspects of communication would further support the claims you make in Table 3 by defining your ten key objectives. 

Author Response

Response:

Thank you for this important and thoughtful comment. We agree that providing a more detailed description of the direction or value of the identified factors would be helpful and interesting addition to the results of this review. However, due to the amount of the reviewed articles, their different methodologies and outcomes it is not easy to use a simple indicator (such as positive or negative value), which would work sufficiently to describe all the factors. Moreover, the aim of this study was “to identify aspects influencing communication about serious news in children and adolescents diagnosed with cancer” (line 55), using the scoping review approach, which is used to map the key concepts (line 60), rather than providing a detailed analysis of particular studies as usualin systematic reviews (Arksey & O’Malley, 2005). As the aim of our study was to identify all potential influencing factors, some of them are reflected in a number of studies, while other factors have been identified in only one or two papers. Providing a value indicator could be also potentially misleading by interpreting the role of the identified factor as facilitators or barriers which was not the aim of this study and such interpretation would be still limited due to the unbalanced amount of evidence behind particular factors as well as more complex meaning of these factors. In the example mentioned by the reviewer, patients’ presence is an important factor, identified in several studies, with some parents considering this helpful and some parents being in opposition to it. Our ambition was to list factors which should be acknowledged and taken into account during difficult conversations with patients and families. Providing a clear guidance on whether particular aspects are desirable, positive or negative is certainly a valuable and interesting goal for future research, but it would be beyond the scope and limits of this review.

However, we have made several amendments to clarify the focus of this study:

  • we changed the subheading in Table 1 from “Theme group” to “Identified aspects of communication” and provided an extra explanation that this table includes information about which aspects were identified in the reviewed articles with more details about particular factors provided in Table 2 and the results section (line 42-43).

  • In the results section, we added a sentence “Table 2 provides more detailed information about the incidence of identified factors across the participants categories.” (lines 197-198)

  • In the study limitations section, we have added a sentence: “This review aimed to identify the aspects that influence communication about serious news with patients and their families. Further analysis of specific aspects, their value or direction (positive / negative) could provide more information about the specific role of these aspects.” (lines 448-451)

Arksey & O’Malley (2005). Scoping studies: Towards a Methodological Framework. Int J Social Research Methodology, 8(1), p.19-32.

Round 2

Reviewer 2 Report

Appreciate the modifications and clarifications. This helped to clarify the scope of the study as well as the need for continued work around the impact of specific communication aspects going forward. 
